# Public health policies for children and youth with special health care needs in OECD member countries and Brazil: A scoping review protocol

Ângela Cristina Rocha Gimenes[ORCID]*, Elenir Rose Jardim Cury*

Graduate Program in Health and Development in Brazil's Center-West Region, Federal University of Mato Grosso do Sul, Campo Grande, Mato Grosso do Sul, Brazil

* angelagimenes3@gmail.com (ACRG); elenir.cury@ufms.br (ERJC)

## Abstract

### Background

Because public health policies lay down guidelines for health promotion in specific populations, a review of policies devised for children and youth with special health care needs (CYSHCN) can reveal the actual degree of priority assigned to this population segment, while also highlighting relevant policies in this field.

### Objective

To map the available evidence of public health policies for CYSHCN in member countries of the Organization for Economic Cooperation and Development (OECD) and Brazil.

### Method

A scoping review protocol was developed as per the Joanna Briggs Institute (JBI) manual and the Preferred Reporting Items for Systematic Review and Meta-Analyses–Extension for Scoping Reviews (PRISMA-ScR) checklist, for application to the Web of Science (WoS), Scopus, PubMed, and Embase databases; to the Latin American and Caribbean Literature in Health Sciences (LILACS) multilingual thesaurus; and to gray literature. The review will map the principal documents (irrespective of time frame or language) addressing public health policies for CYSHCN up to 19 years old. The research protocol has been registered on the Open Science Framework platform (identifier 10.17605/OSF.IO/UW5BH:DOI).

### Results and conclusion

Tables, maps, charts, and/or graphs accompanied by descriptive texts will be employed to present the results to the research question and provide answers to it. Use of both open-access software IRaMuTeQ for similitude analysis and word clouds is also planned. This protocol is expected to reveal policies that meet the specific needs of this vulnerable population segment and highlight examples of good practices or current gaps.

**Data Availability Statement:** No datasets were generated or analysed during the current study. All

relevant data from this study will be made available upon study completion.

**Funding:** This research was supported by funds from Coordenação de Aperfeiçoamento de Pessoal de Nível Superior (CAPES) to ERJC [001].

**Competing interests:** The authors have declared that no competing interests exist.

## Introduction

Public health policies constitute social-protection strategies—in the form of governmental plans, programs, and projects—to minimize social inequality and injustice. They are formulated to meet the demands of populations and social movements for improved living and working conditions [1–4].

The concept of children with special health care needs was coined by the Maternal and Child Health Bureau, in the United States, to refer to children at greater risk of experiencing or developing physical problems, chronic conditions, or problematic behavioral or emotional traits—a population segment whose need for health services and specialized care is greater than that of other children [5].

With the subsequent inclusion of adolescents in the concept, the term was expanded to children and youth with special health care needs (CYSHCN)—an important move, since adolescence is the phase of life when future adults need to prepare to take responsibility for their own health care, as well as for their personal and professional lives [6].

However, a number of gaps remain in policies formulated for CYSHCN. In Brazil, the absence of specific policies for this population segment numbers among these gaps. Also lacking, in broader terms, is a more precise definition of special health care needs, as well as more suitable categorization of these needs to prevent the occurrence of conceptual and epistemological limitations that may hinder access of this population segment to health care services. Furthermore, different terms, such as chronic illness or chronic health condition, have been employed to refer to special needs [7].

Given the recency of the definition of CYSHCN, and the absence of specific Brazilian policies, the review sought to include data from developed OECD member countries and Brazil and compare how these nations have been designing policies for this population. Mapping health policies using the scoping review method involves an initial identification of policy gaps. To this end, a search of both scoping reviews and systematic reviews on the topic of health policies was performed to ensure that the research question had not been previously answered. The databases employed for the search were Portal Periódicos CAPES, Cochrane Library, JBI, Research Registry, and Prospero, using terms and keywords related to the topic. This initial search retrieved no previous reviews or protocols.

Given the breadth of the research question, the scoping review corpus will not be limited to scientific articles, but will include any untrimmed, open-access sources addressing public health policies for CYSHCN, irrespective of language or time frame.

Identifying, tracking, and mapping evidence of public health policies can reveal models, experiences, and contexts, providing support for improvement and reorientation of health care systems and services for CYSHCN and their families.

The purpose of the investigation is to map the evidence on public health policies for CYSHCN available in OECD countries and Brazil.

## Method

A scoping review mapping policies and identifying the main documents and statements, whether from governmental bodies or professional organizations, that are capable of influencing and guiding the nature of practices in this area, will be conducted [8].

### Data collection and analysis

**Scoping review guidelines.**   The review will observe the guidelines proposed by JBI and the recommendations contained in the updated 2021 version of the Preferred Reporting Items

for Systematic Reviews and Meta-Analyses Extension for Scoping Reviews (PRISMA-ScR) [9,10].

**Protocol and registration.** The research protocol has been registered on the Open Science Framework platform (identifier 10.17605/OSF.IO/UW5BH:DOI).

**Research question.** To address these inconsistencies, a scoping review is proposed based on a research question designed under the Population–Concept–Context (PCC) mnemonic, where CYSHCN constitute the population, public health policies are the concept, and OECD member countries and Brazil represent the context. The research question is: What evidence of public health policies for CYSHCN is available in OECD member countries and Brazil?

Two sub-questions were formulated for a better grasp of the topic:

a. What guidelines are orienting practices for CYSHCN?

b. What types of care or treatment are being offered to CYSHCN?

## Eligibility criteria

Scientific articles and other types of literature will be included if containing: (1) information on programmatic provision of public health, defined in policies, plans, interventions, laws, decrees, guidelines, guides, manuals, resolutions from professional boards and associations, consensus statements, clinical recommendations, care provision strategies, or specific lines of care for CYSHCN in OECD member countries and Brazil, consistent with the PCC mnemonic, as outlined above; and (2) elements that respond to the research question. Authors will be contacted if further information on the contents of these sources is required.

Scientific articles and other types of literature addressing health promotion, prevention, early intervention, transition, and palliative care policies; state- or city-level policies.

The age range of CYSHCN (0–19 years) incorporates the World Health Organization (WHO) definition of adolescence as the period between childhood and adulthood, spanning from age 10 to 19 years [11].

For the present study, health policies were defined according to the WHO, as decisions, plans, and actions that are undertaken to achieve specific health care goals within a society. Because this definition is narrowly focused on health care, it may fail to include broader policies that may have an impact on determinants of health that are more in keeping with the health promotion concept [12].

*Information sources and search strategy*. The search strategy employed controlled vocabulary, combining twodatabases—namely, Medical Subject Headings (MESH),for PUBMED, and Embase Subject Headings (EMTREE),for EMBASE—and the Health Sciences Descriptors (DeCS) multilingual thesaurus(for LILACS). Free terms were also used to expand the results and obtain a more sensitive, wider-reaching strategyfor the WoS and Scopus databases.

Designing a sensitive search equation involved consultation with a librarian experienced in database searches in the health sciences area. To prevent possible search errors, the Peer Review of Electronic Search Strategies (PRESS) guideline checklist [13] was employed S1 Fig (submitted 12 Dec., 2022; returned 17 Dec., 2022).

Sensitive search strategies have been deposited in public data repositories:Embase– https://doi.org/10.1079/searchRxiv.2023.00314, SCOPUS– https://doi.org/10.1079/searchRxiv.2023.00315, Web of Science - https://doi.org/10.1079/searchRxiv.2023.00316, LILACS - https://doi.org/10.1079/searchRxiv.2023.00317, PUBMED - https://www.cabidigitallibrary.org/doi/10.1079/searchRxiv.2023.00307.

The search strategy will be applied to databases, but will also entail manual search of specific websites–*e.g.*, the WHO, the United Nations Children's Fund (UNICEF), the Guidelines International Network, the US National Guideline Clearinghouse, Google Scholar, the Catalog of Theses and Dissertations of the Coordination for the Improvement of Higher Education Personnel (CNPq, Brazil), the Brazilian Digital Library of Theses and Dissertations, Open Access Theses and Dissertations (OATD), ProQuest Dissertations & Theses Global (PQDT), and the Open Access Scientific Repositories of Portugal (RCAAP), as well as specialized websites (pediatric associations) and all the websites of Ministries of Health (or their equivalents) in OECD member countries and Brazil.

Development of the search strategy will follow the steps outlined in Fig 1.

The search strategy involved three steps, drawing on JBI principles [9]. The first step entailed searching the Medical Literature Analysis and Retrieval System Online (MEDLINE)

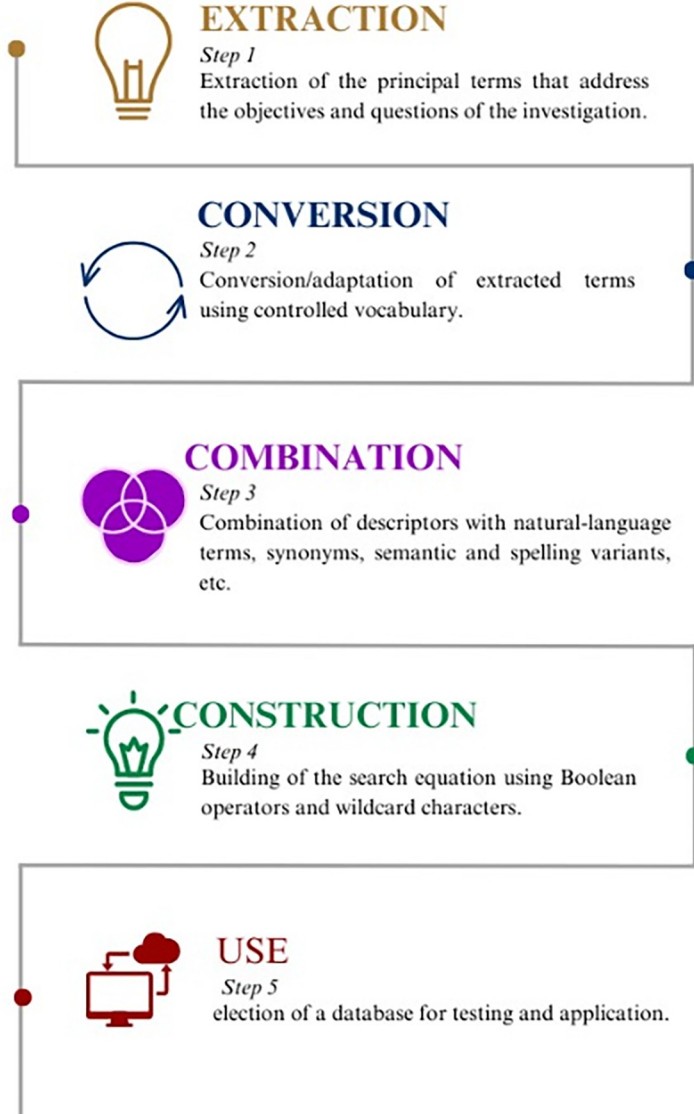

**Fig 1. Steps for developing a search strategy for databases [14].**

and LILACS for the largest possible number of terms and keywords related to the topic, so as to design the search string. In step 2, the search equation was applied to the databases on 22 Dec., 2022. In step 3, additional sources (*e.g.*, reference lists in articles or reports retrieved) will be searched.

Since the CYSHCN-related terminology was established for planning and developing large programs, it has a broad scope, and the search equation based on this terminology yielded a high number of scientific articles, as described below.

The search strategy was applied to fourdatabases: WoS (334 results), PubMed (9934), Embase (981), and Scopus (1094), as well as to theDeCSthesaurus (172 results).

## Selection of sources of evidence

The 12,515 records retrieved were imported into Rayyan software [15] (Qatar Computing Research Institute) and 739 duplicates were thus identified and removed, resulting in 11,776 articles. The next step will entail having this output analyzed by two independent reviewers, who will grade the titles and abstracts for relevance to the topic (into three categories: 'yes', 'no,' 'possibly') and subsequently read the full texts. Disagreements will be resolved through consensus-based discussion. Sources identified as relevant in the citation lists of the selected studies will also be retrieved and analyzed.

A calibrating, pilot inter-reviewer test will be carried out with 40items of evidence. A minimum 75% agreement rate will be required for the scoping review to proceed.

The screening process will be depicted as a PRISMA-ScR flowchart showing selected, duplicate, and removed records(with reasons for exclusion), as well as any additions from the third selection step (Fig 2).

## Data extraction process

After selection of the articles that will comprise the study sample, a Microsoft Office Excel spreadsheet [16] adapted from the JBI method, will be prepared, containing the following fields: authors; year of publication; origin/country; goals; method; health policies; related

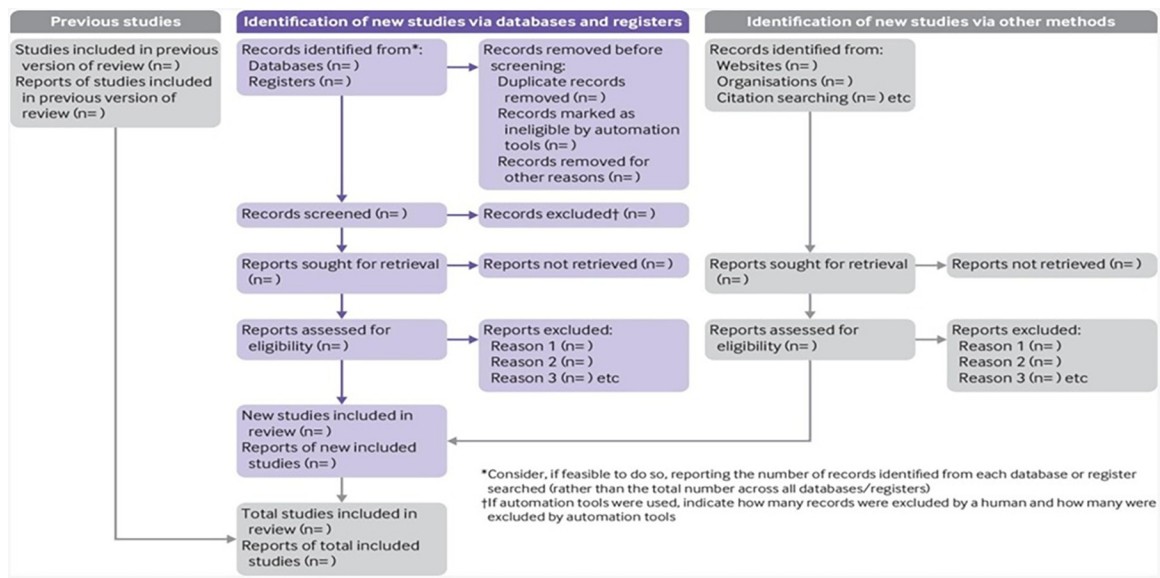

**Fig 2. PRISMA-ScR 2020 [10] flowchart template.**

terms used; categorizations of CYSHCN; official documents cited; types of care provided. A descriptive text will report the health policies found in the review, together with respective authors and other data relevant for responding to the research question. A summary of the principal results will provide an overview of health policies, related terms, categorizations, knowledge gaps encountered, and relevance to further studies and society.

The data extraction form will be pilot-tested by two team members on three sources, with one reviewer extracting data and the other checking that all relevant results have been extracted.

## Summary of evidence

Tables, maps, charts, and/or graphs combined with descriptive texts will be generated to present the results to the research question. Use of both open-access software IRaMuTeQ [17] for similitude analysis and word clouds is also planned.

## Ethical issues

This study entails a scoping review and therefore approval from a research ethics committee is waived. However, resolution 466, of 2012, issued by the Brazilian Health Council (CNS) [18] will be complied with as regards the analysis and sharing of results.

## Limitations

Expansion of the search equation may not prevent some articles from being missed due to indexing errors. To minimize these losses, a librarian experienced in database searches in the health sciences area will be consulted. Choice of descriptors may also lead to loss of potentially useful articles.

## Relevance, economic viability, and feasibility of the project

Given that health care provision to CYSHCN is a recent topic, the findings of a scoping review can potentially broaden the discussion on the subject, warranting the design of this search protocol. The project does not require significant financial resources and can be privately funded.

## Supporting information

**S1 Checklist. PRISMA-P 2015 checklist.**
(DOCX)

**S1 Fig. PRESS guideline–search submission and peer review assessment.**
(TIF)

**S2 Fig. Peer Review of Electronic Search Strategies (PRESS) guideline checklist.**
(DOCX)

## Acknowledgments

The authors wish to thank Professor Samuel Miranda Mattos and librarian Camila Belo for their kind assistance.

## Author Contributions

**Conceptualization:** Ângela Cristina Rocha Gimenes.

**Investigation:** Elenir Rose Jardim Cury.

**Methodology:** Ângela Cristina Rocha Gimenes.

**Supervision:** Elenir Rose Jardim Cury.

**Validation:** Elenir Rose Jardim Cury.

**Visualization:** Ângela Cristina Rocha Gimenes.

**Writing – original draft:** Ângela Cristina Rocha Gimenes.

**Writing – review & editing:** Elenir Rose Jardim Cury.

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
