## [Decision Letter · Decision Letter 0]

19 Jul 2023

PONE-D-23-16966Public health policies for children and youth with special health care needs in OECD member countries and Brazil: a scoping review protocol

PLOS ONE

Dear Dr. GIMENES,

Thank you for submitting your manuscript to PLOS ONE. After careful consideration, we feel that it has merit but does not fully meet PLOS ONE’s publication criteria as it currently stands. Therefore, we invite you to submit a revised version of the manuscript that addresses the points raised during the review process.

Dear Authors, I am writing because your manuscript submitted to PLOS ONE has been reviewed. Please find the reviewer comments at the bottom of this email.

Since the reviewers do find merit in the paper, we would be willing to reconsider if you wish to undertake **minor revisions and resubmit**, fully addressing the referees’ concerns enumerated below.

Please be as specific as possible in your response to the reviewers but do not include any author contact information and/or names as this will be shared with the reviewers and it is important to keep the review process anonymous.

We look forward to receiving your revised manuscript.

Kind regards,

Ana Larissa Gomes Machado, Ph.D

Academic Editor

PLOS ONE

Reviewers' comments:

Reviewer's Responses to Questions

**Comments to the Author**

1. Does the manuscript provide a valid rationale for the proposed study, with clearly identified and justified research questions?

Reviewer #1: Partly

Reviewer #2: Yes

2. Is the protocol technically sound and planned in a manner that will lead to a meaningful outcome and allow testing the stated hypotheses?

Reviewer #1: Yes

Reviewer #2: Yes

3. Is the methodology feasible and described in sufficient detail to allow the work to be replicable?

Reviewer #1: Yes

Reviewer #2: Yes

4. Have the authors described where all data underlying the findings will be made available when the study is complete?

Reviewer #1: No

Reviewer #2: Yes

5. Is the manuscript presented in an intelligible fashion and written in standard English?

Reviewer #1: Yes

Reviewer #2: Yes

6. Review Comments to the Author

You may also provide optional suggestions and comments to authors that they might find helpful in planning their study.

Reviewer #1: study scope review protocol. Study suitable for publication. It presents an introduction, consistent method.

Reviewer #2: I suggest removing the guiding question from the objective and adapting the objective as follows: mapping the evidence about public health policies available for children and young people with special health needs in OCDE countries and in Brazil.

Correct in the abstract and in the text that LILACS is a bibliographic index and not a database. Add in the abstract that will include the gray literature.

Remove articles not available in full as an exclusion criterion. In view of the numerous resources currently available to retrieve articles, it is not justified to use this exclusion criterion.

Specify which controlled vocabulary you used for each database or index (Example: LILACS (DeCS), MesH (PubMed)...

Which database did you use Thesaurus on? To specify.

I suggest removing the description of the PCC Mnemonic as well as the research question from the introduction and adding it to the methodology. Create a subtopic (guiding question or research question) after the registration protocol and describe the PCC Mnemonic and the guiding question.

Will you use software such as EndNote for removing duplicates, and Rayyan for blindly peer-reviewing studies? If yes, add. If not, describe how this systematic selection of studies will be carried out manually.

7. PLOS authors have the option to publish the peer review history of their article (what does this mean?). If published, this will include your full peer review and any attached files.

Reviewer #1: **Yes: **Dr. Samuel Miranda Mattos

Reviewer #2: **Yes: **Jefferson Abraão Caetano Lira

---

## [Author Response · Author response to Decision Letter 0]

30 Jul 2023

Dear Reviewers:

The Reviewers’ remarks were addressed as follows:

1) The principal objective has been rephrased.

2) Both the abstract and the main text now inform that DeCS (LILACS) is a multilingual thesaurus (the term employed on the DeCS website to describe itself).

3) Mention to the gray literature has been added to the abstract.

4) Articles not being available in full was removed as an exclusion criterion.

5) The controlled vocabulariesused for each database and index have now been specified.

6) The suggestion not to mention the PCC Mnemonic in the Introduction and research question was accepted. This information has now been added to the Methods section.

7) Use of Rayyan software for identification and exclusion of duplicates has now been mentioned.

Sincerely, 

The authors

---

## [Editor Report · Decision Letter 1]

29 Aug 2023

Public health policies for children and youth with special health care needs in OECD member countries and Brazil: a scoping review protocol

PONE-D-23-16966R1

Dear Dr. GIMENES,

We’re pleased to inform you that your manuscript has been judged scientifically suitable for publication and will be formally accepted for publication once it meets all outstanding technical requirements.

Kind regards,

Ana Larissa Gomes Machado, Ph.D

Academic Editor

PLOS ONE

---

## [Editor Report · Acceptance letter]

2 Oct 2023

PONE-D-23-16966R1 

Public health policies for children and youth with special health care needs in OECD member countries and Brazil: A scoping review protocol 

Dear Dr. Gimenes:

I'm pleased to inform you that your manuscript has been deemed suitable for publication in PLOS ONE. Congratulations! Your manuscript is now with our production department. 

Kind regards, 

on behalf of

Dr. Ana Larissa Gomes Machado 

Academic Editor

PLOS ONE